# Alterations in Measures of Body Composition, Neuromuscular Performance, Hormonal Levels, Physiological Adaptations, and Psychometric Outcomes during Preparation for Physique Competition: A Systematic Review of Case Studies

**DOI:** 10.3390/jfmk8020059

**Published:** 2023-05-08

**Authors:** Brad J. Schoenfeld, Patroklos Androulakis-Korakakis, Alec Piñero, Ryan Burke, Max Coleman, Adam E. Mohan, Guillermo Escalante, Alexa Rukstela, Bill Campbell, Eric Helms

**Affiliations:** 1Department of Exercise Science and Recreation, Lehman College, City University of New York, Bronx, NY 10468, USA; polkarots@gmail.com (P.A.-K.); rburke3320@gmail.com (R.B.); colemanmax888@gmail.com (M.C.); adam.mohan@lc.cuny.edu (A.E.M.); 2Department of Kinesiology, California State University San Bernardino, San Bernardino, CA 92407, USA; gescalan@csusb.edu; 3Exercise Science Program, University of South Florida, Tampa, FL 33620, USA; arukstela@usf.edu (A.R.); bcampbell@usf.edu (B.C.); 4Sport Performance Research Institute New Zealand (SPRINZ), Auckland University of Technology, Auckland 1010, New Zealand; eric.helms@aut.ac.nz; 5Muscle Physiology Laboratory, Department of Exercise Science and Health Promotion, Florida Atlantic University, Boca Raton, FL 33431, USA

**Keywords:** bodybuilding, bikini, figure, pre-contest, effects

## Abstract

The present paper aimed to systematically review case studies on physique athletes to evaluate longitudinal changes in measures of body composition, neuromuscular performance, chronic hormonal levels, physiological adaptations, and psychometric outcomes during pre-contest preparation. We included studies that (1) were classified as case studies involving physique athletes during the pre-contest phase of their competitive cycle; (2) involved adults (18+ years of age) as participants; (3) were published in an English-language peer-reviewed journal; (4) had a pre-contest duration of at least 3 months; (5) reported changes across contest preparation relating to measures of body composition (fat mass, lean mass, and bone mineral density), neuromuscular performance (strength and power), chronic hormonal levels (testosterone, estrogen, cortisol, leptin, and ghrelin), physiological adaptations (maximal aerobic capacity, resting energy expenditure, heart rate, blood pressure, menstrual function, and sleep quality), and/or psychometric outcomes (mood states and food desire). Our review ultimately included 11 case studies comprising 15 ostensibly drug-free athletes (male = 8, female = 7) who competed in various physique-oriented divisions including bodybuilding, figure, and bikini. The results indicated marked alterations across the array of analyzed outcomes, sometimes with high inter-individual variability and divergent sex-specific responses. The complexities and implications of these findings are discussed herein.

## 1. Introduction

Physique competition is a sport in which athletes are judged on a combination of muscle size, appearance of low body fat (BF), overall symmetry, and presentation [1,2]. The competitive cycle for physique athletes consists of an “off-season” that customarily includes a caloric surplus and resistance training (RT) with the goal of increasing lean body mass while minimizing BF accrual, as well as an “in-season” preparatory phase (sometimes referred to as a ‘contest preparation phase’), which typically consists of caloric restriction, RT, and aerobic activity to maintain lean body mass and significantly decrease BF [3,4]. There are also shorter phases including the peak week, which includes the manipulation of variables in the final week before competition, as well as the recovery phase, where athletes methodically regain BF [5,6]. Throughout all the phases, athletes may manipulate their RT, nutritional intake, hydration status, aerobic activity, supplementation, and in some instances utilize performance- and image-enhancing drugs (PIEDs) [7].

Numerous case reports have followed competitive physique athletes along their competition preparatory phases. These preparatory phases are of particular interest owing to the tremendous amount of physiological and psychological stress they place on the athlete, as the methods of achieving a winning physique closely mimic a controlled starvation state. Previous research has reviewed the impact of chronic energy deficits, noting impaired physiological functioning when energy availability falls below a certain threshold for an extended time period [8]. Historic research has also documented the detrimental effects of severe caloric restriction on metabolic rate, mental well-being, sociability, and health markers [9].

While energy restriction of this magnitude for extended periods of time is atypical of today’s competitive environment outside of physique sport, physique athletes willingly engage in behaviors that mimic semi-starvation for defined periods of time. In this regard, they accept the associated potentially negative outcomes as part of their sport. This provides researchers with a unique opportunity to study the mind and body in this state, which has been best documented through observational case reports. The present paper aimed to systematically review case studies on physique athletes to evaluate longitudinal changes in measures of body composition, neuromuscular performance, chronic hormonal levels, physiological adaptations, and psychometric outcomes during pre-contest preparation. The findings provide insights into the chronic responses to preparation for physique contests and hopefully will inspire research into better practices for athletes to optimize results while minimizing negative effects.

## 2. Methods

### 2.1. Literature Search

We preregistered the methods for this review prior to data collection on the Open Science Framework website (https://osf.io/d67qf, accessed on 1 February 2023). We searched the PubMed/MEDLINE, Scopus, and Web of Science databases from inception to February 2023 to locate studies relevant to the topic of interest. Two researchers (R.B. and A.P.) screened the abstracts and reviewed the full texts for studies that conceivably met inclusion criteria. Inclusion required agreement between both researchers; in cases where a disagreement arose, a third researcher (B.J.S.) resolved the dispute.

We employed the following Boolean syntax for carrying out the search: (“case study” OR “case series”) AND (“bodybuilder” OR “body-builder” OR “body builder” OR “bodybuilding” OR “physique athlete” OR “physique competitor” OR “bikini athlete” OR “bikini competitor” OR “figure athlete” OR “figure competitor”) AND (“competition” OR “contest”). Other case studies included in our paper were either known by the authors or discovered by searching through the reference sections of retrieved articles. Our methods adhered to the guidelines set forth by the Preferred Reporting Items for Systematic Reviews and Meta-Analyses (PRISMA) [10].

### 2.2. Inclusion/Exclusion Criteria

We included studies that (1) were classified as case studies involving physique athletes during the pre-contest phase of their competitive cycle; (2) involved adults (18+ years of age) as participants; (3) were published in an English-language peer-reviewed journal; (4) had a pre-contest duration of at least 3 months; (5) reported changes across contest preparation relating to measures of at least one of the following variables: body composition (fat mass, lean mass, and bone mineral density), neuromuscular performance (strength and power), chronic hormonal levels (testosterone, estrogen, cortisol, leptin, and ghrelin), physiological adaptations (maximal aerobic capacity, resting energy expenditure, heart rate (HR), blood pressure (BP), menstrual function, and sleep quality), and/or psychometric outcomes (mood states and food desire). Studies were excluded if (1) they were a poster or oral presentation at a conference, or if they were an unpublished thesis or dissertation; and (2) participants had existing injuries, medical conditions, or disabilities.

### 2.3. Data Coding and Analysis

As previously described [11], data were extracted from the respective studies and coded in an Excel spreadsheet (Microsoft Corporation, Redmond, Washington) by two authors (A.P. and P.A.K.) using the following classifications: (1) study characteristics (author and year of publication); (2) participant demographics (age, sex, and RT experience); (3) study methods (type of test and outcome assessed); and (4) numerical values for each outcome at each time point. In cases where studies lacked sufficient information regarding pre–post changes, we contacted the authors to request the missing data. If we were unable to acquire data from authors, we extracted values from figures using WebPlotDigitizer online software (https://apps.automeris.io/wpd/ (accessed on 3 May 2023)) where applicable. To account for the possibility of coder drift, a third researcher (A.M.) recoded ~30% of the studies, which were randomly selected for assessment [12]. Per case agreement was determined by dividing the number of variables coded the same by the total number of variables. Acceptance required a mean agreement of ≥90%. Any discrepancies in the extracted data were resolved through discussion and mutual consensus of the coders.

We calculated measures of central tendency to elucidate the effects on outcomes. When multiple methods were employed to assess a given outcome (e.g., body composition), we averaged the results of the methods to derive a final value. We interpreted data based on estimations of the magnitude of changes over time.

## 3. Results and Discussion

A total of 12 studies met the initial inclusion criteria. However, we were not able to obtain the necessary information for 1 study [13]; thus, our review included 11 case studies comprising 15 athletes (male = 8; female = 7). Figure 1 provides a PRISMA flow chart of the search process; Table 1 provides descriptive information of the included studies.

Competition experience of male athletes ranged from having only previously competed in a regional qualifier to a professional athlete. Competition experience of female athletes ranged from preparing for the first competition to having previously placed in the top 5 at a national competition. The age of athletes ranged from 21 to 44 years; nine athletes were in their twenties, four athletes were in their thirties, and two athletes were in their forties. All athletes either tested negative for anabolic agents prior to their contest or claimed to be drug-free. The study periods ranged from 14 to 32 weeks.

### 3.1. Resistance Training

Seven athletes did not report RT volume. Of those who did, four athletes reported completing 4 to 10 sets per muscle group per week; one reported 10 to 20 sets per muscle group per week; one reported 12 to 24 sets per muscle group per week; one reported 16 to 20 sets per muscle group per week; and, while one did not report sets per muscle group, they reported a range of 138 to 207 total sets per week for all muscle groups combined. All athletes reported their RT training frequency, with 14 reporting training at least 4 days per week, 7 reporting training at least 5 days per week, 1 reporting training at least 6 days per week, and 1 reporting training from 3 to 7 days per week. Only three athletes reported relative RT intensity: one reported training to failure on all sets, one reported training between 1 repetition in reserve (RIR) and failure, and one reported training between 1 and 3 RIR, respectively.

When stratified by sex, five males reported RT volume. Three of them were between 4 and 10 sets/muscle grp/week (all from Chappell et al., so conceivably on a similar program). The other two used 16–20/week and 12–24/week. All males reported RT frequency, with seven training at least 4×/week and one training 3–7×/week. Alternatively, three females reported RT volume: one using 4–10 sets/muscle grp/week, one using 10–20 sets/muscle grp/week, and one only reported total number of sets 138 to 207 sets/week. All females reported RT training at least 4×/week, with none reporting training more than 6×/week.

### 3.2. Aerobic Training

All athletes reported engaging in some form of aerobic training (AT) leading up to their respective competitions, with the number of weekly sessions ranging from 0 to 16 and all but four reporting ≤ 7 training sessions per week. Five athletes reported increasing AT frequency as their competition approached. Aerobic training time was often reported as a range, making it difficult to accurately calculate total weekly durations. However, 12 athletes reported engaging in at least 50 min and 11 reported engaging in at most 300 min of AT per week. Only 1 athlete reported exclusive use of high-intensity interval training (HIIT), while 4 athletes reported engaging only in steady state (SS) AT, and the remaining 10 reported using a combination of HIIT and SS. Three athletes reported some form of walking or running as their sole mode of AT; two reported using only a stair climber; and four reported utilizing a combination of two or more of the following modalities: running/walking, stair climber, elliptical, stationary bike, or spin bike. Six athletes did not provide information on their chosen mode of AT.

When stratified by sex, all males reported AT volume, with six reporting at least 40 min/week, one reporting as few as 0 min/week, and one as few as 30 min/week. Actual volume was difficult to calculate as most gave a weekly frequency range and a single-session duration range. All males reported an AT frequency, with six training at least 2×/week, one reporting as few as 0×/week, and one reporting as few as 1×/week. Four reported maximal frequencies of 7×/week or greater, with three reporting AT frequency of up to 13×+/week. Alternatively, all females reported an AT volume of at least 50 min/week. Actual volume was difficult to calculate as most gave a weekly frequency range and a single-session duration range, although two of the three that gave an estimation of the total volume reported less than 200 min of AT/week. All females reported AT frequency, with six training at least 3×/week and one reporting as few as 1×/week; only one reported a frequency of greater than 6×/week.

### 3.3. Posing Practice

Two athletes, both male, reported practicing posing during the duration of their respective preparation. One practiced two to four times per week beginning 6 weeks prior to competition, while the other increased in frequency from one time per week 26 weeks out from competition to three to four times per week beginning 6 weeks prior to competition, with posing sessions lasting from 15 to 30 min each.

### 3.4. Caloric and Protein Intake

Initial dietary intake data were recorded for all competitors at the beginning of their respective preparatory phases, and final data were recorded prior to their respective competitions. The timing of final intake data across studies was heterogeneous, ranging from ≥2 weeks away from competition to during peak week. Absolute initial and pre-contest daily caloric intake ranged from 1306 kcal to 2500 kcal at initial measurement and from 1051 kcal to 2038 kcal for females at the final measurement; for males, initial daily caloric intake ranged from 2155 kcal to 3860 kcal and final caloric intake ranged from 1548 kcal to 2855 kcal. The largest individual initial-to-final differences for females and males were 1189 kcal and 2136 kcal, respectively, while the smallest absolute differences were 255 kcal and 52 kcal, respectively.

In general, daily caloric intake tended to decrease from initial to final data collection, with only two athletes reporting higher final than initial caloric intakes. One of these athletes was female and the other male, and they reported respective absolute increases of 361 kcal and 52 kcal per day, with relative increases of 10 kcal/kg bodyweight/day and 5 kcal/kg bodyweight/day, respectively. Although the final daily caloric intake of most athletes was lower than their initial respective intakes, eight athletes reported their lowest energy intake at some point between their initial and final values, while five reported their final daily intake as their lowest, and two did not provide intake information between their initial and final measurements.

Protein intake ranged from 2.2 to 3.4 g/kg/day for females and 2.1 to 3.6 g/kg/day for males. Initial mean protein intake was 2.6 g/kg/day for females and 2.7 g/kg/day for males. Final mean protein intake was 2.9 g/kg/day for females and 2.8 g/kg/day for males.

The purpose of this review was to provide insights into chronic changes that occur during pre-contest preparation for physique competition as objectively assessed in a laboratory setting. The results indicated marked alterations across the array of analyzed outcomes, sometimes with high inter-individual variability and divergent sex-specific responses. What follows is a discussion of these observed alterations and their potential implications.

### 3.5. Body Composition Changes

All athletes included in this review experienced substantial decreases in fat mass over the course of their pre-contest period. Mean reductions in BF were similar between the sexes (8.7% and 8.2% for males and females, respectively). Fat loss patterns were generally linear, with progressive decreases observed throughout the pre-contest period in the majority of athletes irrespective of sex. The rate of weight loss was relatively gradual across the pre-contest period, averaging −0.36 kg/week for all athletes. Overall, female athletes displayed modestly slower absolute rates of weight loss compared with males (−0.28 vs. −0.42 kg/week, respectively).

The female athletes started their pre-contest preparation with a substantially higher %BF compared with the males (21.3% vs. 14.5% respectively) and competed at a substantially higher %BF than their male counterparts as well (13.1% vs. 5.8%, respectively). Only one female athlete competed at a single digit %BF [14], while all male athletes had BF levels of less than 10%. This is consistent with judging criteria for the studied physique contest divisions: all male athletes in our review competed in men’s bodybuilding, which rewards athletes for having extremely low levels of BF, whereas all female athletes competed in figure or bikini divisions, which sometimes penalize excessive leanness [1].

Changes in lean mass varied between athletes. These variances were generally sex-specific, with female athletes either maintaining or slightly increasing lean mass, while male athletes tended to show a loss of lean mass. A possible explanation for this sex-related discrepancy is the somewhat more rapid weight loss experienced by the male athletes. Evidence indicates that a slower rate of weight loss is conducive to maintaining lean mass [24,25]. However, the absolute amount of weekly weight loss experienced by the male athletes (−0.42 kg/week) was consistent with that shown to preserve lean mass in elite athletes [25], calling into question whether the rate of loss was an explanatory factor. It should be noted that the athletes in the aforementioned intervention [25] were not physique athletes and had a mean post-study BF level of 15.5%. Accordingly, an alternative explanation could be that female athletes had higher amounts of fat mass at the end of contest preparation compared with males, with almost all displaying double-digit %BF levels immediately prior to competition. Although speculative, it is conceivable that lean mass losses accelerate when %BF approaches essential levels, which might have occurred more often in the male athletes competing in a division where extreme leanness is rewarded. This phenomenon of lean mass loss occurred even with a relatively high dietary protein intake, as the male athletes in the included studies consumed an average of 2.8 g/kg/d, which is within the range of current recommendations for optimizing preservation of lean mass during an energy deficit [26]. We note that women have higher levels of essential fat compared with men, which must be considered in the context of the observed results. Thus, this hypothesis warrants further study.

Given that energy status may affect bone mass [27], it is pertinent to explore whether pre-contest preparation may be detrimental to bone mineral density (BMD). The terms relative energy deficiency in sport (RED-S) and the female athlete triad have been coined to describe the effect of energy status on health outcomes, including BMD. Although both syndromes have commonalities, they differ somewhat in their scope of outcomes and target populations [28]. Of note, RED-S takes a broad-based perspective on the topic that encompasses both sexes while the triad is specific to the effects of energy status on girls and women. The sex-related distinction between terms is relevant to changes in BMD, as low energy availability has particularly detrimental effects on females in this regard [29].

Despite the well-established health-related issues with low energy availability, none of the three studies that estimated changes in BMD showed a loss of bone mass during pre-contest preparation [15,16,18]. The studies comprised a total of four athletes (one male, three females) and used DXA to calculate BMD, which is considered the gold standard for assessment [30]. On the surface, this would suggest that pre-contest preparation strategies, as commonly practiced, do not negatively impact bone health. However, it is not clear if repeated periods of pre-contest preparation may negatively affect bone mass over time. Future research should thus seek to assess BMD status in individuals who have competed in multiple physique contests. It is conceivable that the athletes’ regimented RT programs offset any negative effects of low energy status on bone health [31]. This hypothesis requires further investigation.

### 3.6. Hormonal Levels

Evidence suggests that a prolonged energy deficit, combined with high levels of exercise and physical activity, can affect anabolic and catabolic hormone levels. In this review, five studies (*n* = 8; male = 6; female = 2) reported on testosterone [5,17,20,21,22]; four reported on leptin (*n* = 5; male = 2; female = 3) [16,17,20,21]; three reported on ghrelin (*n* = 4, male = 2; female = 2) [16,17,21]; three reported on cortisol (*n* = 3, male = 2; female = 1) [17,20,21]; and two reported on estrogen (*n* = 3, female = 3) [16,20].

Among the six male athletes whose testosterone levels were measured, levels generally dropped in an almost linear fashion throughout the contest preparation. Baseline values ranged from 543 to 922 ng/dL, and all fell to hypogonadal levels (175 to 227 ng/dL) toward the end of their respective competition preparation periods. Conversely, in females, testosterone levels tended to increase or remain relatively stable over the course of the contest preparation phase. Interestingly, it has been reported that weight loss in general reduces testosterone concentrations [32,33]. Although the testosterone levels of females in the included case studies increased or remained stable, the authors of another investigation observed a decrease in testosterone levels in female fitness competitors during dieting, but noted it may have not been physiologically meaningful because the mean values were still within normal values of serum testosterone [34]. It should be noted, however, that in the investigation with female fitness competitors [34], researchers assessed testosterone levels over a 20-week dieting phase and not a pre-contest phase. As such, it is difficult to make direct comparisons between studies. Although speculative, it is conceivable that testosterone levels in females are less responsive to low energy availability and weight loss.

Estrogen levels were reported in three female athletes. There was considerable variability in the responses of initial estrogen levels to pre-contest. In one athlete, estrogen levels were significantly elevated at the end of the pre-contest period. Conversely, estrogen levels of the other two competitors either slightly dropped or remained relatively stable. During weight loss, estrogen levels typically decline; however, prior research on this topic was conducted in postmenopausal women and may not necessarily reflect what occurs in younger women [33]. Given evidence indicating that estrogen is the primary female anabolic hormone [35], the lack of significant declines in estrogen levels may have contributed to the general preservation of lean tissue observed throughout the preparation period in the female athletes. This contrasts with the male athletes, who universally experienced dramatic drops in testosterone, generally to hypogonadal levels, and concomitantly demonstrated substantial lean tissue losses.

Cortisol levels were reported in three competitors (two male and one female). In all three competitors, cortisol increased as the competition preparation period progressed. The rise in cortisol levels, as has been previously suggested, may partly be due to an increase in lipolysis and (to some extent) muscle tissue proteolysis as competitors undergo a long caloric deficit during contest preparation [21]. Moreover, because calorie intake is limited toward the end of contest preparation and physical activity usually increases as a result of more cardiovascular exercise and posing practice, this may lead to an increase in cortisol levels [36].

Leptin levels were reported in three female and two male athletes. In two of the females, leptin levels remained relatively stable throughout the contest preparation period, but in one female, leptin levels dropped significantly. Similarly, leptin levels remained stable for one male athlete and slightly dropped over time for the other male athlete. The inconsistency in the case studies in this meta-analysis contrast with previous work [37] reporting that leptin levels fall during prolonged exposure to a calorie deficit among bodybuilders preparing for competition. The reason for these discrepancies in findings remains unclear but may involve an interaction between genetic factors and pre-contest preparation strategies. This hypothesis warrants further study.

Ghrelin levels for the participants in this meta-analysis were reported for two female and two male participants. Except for one female participant whose ghrelin concentrations remained relatively stable, levels of this hormone typically increased over time as the contest preparation period progressed. Previous research has supported the notion that ghrelin levels tend to increase throughout the contest preparation period [37], https://paperpile.com/c/B9l3gq/RV8K (accessed on 3 May 2023). As ghrelin is strongly involved in the regulation of energy homeostasis [38], an increase in ghrelin levels after a prolonged caloric deficit is to be expected.

### 3.7. Neuromuscular Performance

Power was assessed via different methods including critical power on a cycle ergometer, the Wingate test, and the vertical jump test. All athletes in studies that assessed power experienced absolute decreases in this outcome over the course of their pre-contest period, though some either maintained or increased their power when expressed relative to their bodyweight [5,20]. The current literature on the effects of lean and fat mass loss on power suggests that the duration of a weight loss intervention may play a crucial role in power performance changes, with longer interventions resulting in greater power decreases [25]. However, there is evidence to suggest that weight loss, even in cases where lean mass is lost, will not necessarily negatively impact power performance, even in highly trained athletes [25].

Additionally, despite losses in absolute power observed when comparing the beginning to the end of the pre-contest period, some athletes experienced increases in absolute power during the early stages of the pre-contest period, which dissipated as the athletes approached their competition and reached body fat levels below ≈20% for women and below ≈10% for men. The observed increases in absolute power during early contest preparation may be the result of the athletes becoming more familiar with the testing modality employed to assess power performance. Overall, the duration of the pre-contest period in conjunction with the substantial losses in bodyweight, low energy availability, and in some cases the loss of lean mass may explain the decreases in absolute power observed. The ability of the athletes to maintain, and in some cases even increase, their relative power may be explained by their consistent engagement in RT. Although it is unclear if some athletes performed power-specific training, ‘hypertrophy-oriented’ RT can also result in power improvements [39].

Five of the 11 studies included assessed strength. Overall, most athletes were able to maintain or increase their strength. Although somewhat surprising, previous research has shown that, similarly to power, strength increases are possible during periods of substantial weight loss, even in trained athletes [25], with recent meta-analytic evidence showing that an energy deficit does not seem to meaningfully affect strength gains, despite its potential to cause lean mass losses [40].

In the lone study where an athlete experienced absolute strength decreases in the squat, bench press, and deadlift exercises, their relative strength remained similar or even slightly increased compared with baseline measures [21]. Bodyweight and lean mass are positively associated with absolute strength in powerlifters and men performing the squat, bench press, and deadlift [41,42], which may explain the decreases observed in this athlete’s absolute strength.

Interestingly, athletes in two of the included studies [5,22] experienced absolute strength increases (assessed via hand grip strength and isometric knee extension). Previous research on weight loss has shown that hand grip strength may be unaffected even in cases where lean mass losses have been observed [43]. The increases in knee extension strength observed in the study by Schoenfeld et al. [22] may be due to the competitor including the leg extension exercise as a regular component of his training regimen, which was similar to the test modality employed (i.e., isometric knee extensions), and thus consistent with the principle of specificity. Alternatively, in the study by Tinsley et al. [23], where strength was assessed as the peak force during the squat exercise, the female competitor experienced increases of up to 5% and 15% and decreases of up to 25% and 9% for concentric and eccentric force, respectively, from initial measurements. Similarly, relative force production increased up to 9% and 19% and decreased up to 20% and 2% for concentric and eccentric force, respectively. Interestingly, the largest drops in both concentric and eccentric relative strength corresponded to the measurement taken between two competitions when the competitor recorded their highest amount of lean mass.

Given that the absolute power and strength changes observed pre-contest may have been impacted by lean mass losses and low energy availability, they could be assumed to be temporary. In addition to exploring the effects of the pre-contest period on strength and power, future research should investigate how strength and power respond to the post-contest period where energy availability is higher and when athletes usually regain some of the lean mass they lost in the weeks leading up to the competition.

### 3.8. Physiological Adaptations

The physiological adaptations to contest preparation such as changes in menstrual function, cardiovascular function, resting metabolic rate (RMR), and sleep quality followed similar general trends among athletes. Sleep quality was only discussed in two studies; in both cases, sleep was negatively impacted to some degree [17,18]. As reported by Petrizzo et al. [18], despite actigraphy showing no changes in sleep quality, Pittsburgh Sleep Quality Index scores were reasonably high (indicating poor sleep quality) during contest preparation. Further, Pardue et al. [17] reported that the athlete experienced some bouts of insomnia during contest preparation. With sleep-related data only reported for two athletes, more research is needed to determine the commonality of these experiences.

Among the four female athletes included in this review for whom menstrual cycle data were reported, three experienced amenorrhea, either early on [23], mid-way [14], or late [18] in contest preparation, with the final female who did not experience amenorrhea experiencing delayed menstruation by 4 and 5 days in her final three recorded menses, respectively [20]. Cardiovascular changes were also commonly observed, as BP decreased in 8 of the 10 athletes in whom it was measured from small to large magnitudes (−2 to −28 mmHg systolic; −4 to −30 mmHg diastolic), and HR decreased in 9 of 10 athletes (−1 to −26 bpm). Changes in VO_2_ max were recorded for four athletes and varied, with an increase in one athlete [15], a decrease in another [44], and an initial increase followed by a decrease later in preparation in two others [20,21]. While RMR decreased in 9 of 11 athletes from the start to the end of contest preparation, this is to be expected with weight loss [45]. Further, some end-of-preparation measurements were possibly confounded by acute increases in energy intake associated with peak week [5].

While many of these physiological changes were reasonably consistent, the causative factors are potentially multifaceted. Physique athletes not only increase their AT volume (and sometimes RT as well) during contest preparation, but they also restrict energy intake and, consequently (as a goal), achieve very low BF levels; each of these factors can independently impact physiology [46,47,48]. For example, bradycardia, insomnia, and poor sleep quality are symptoms of overreaching [46]; menstrual cycle disruption is a repercussion of RED-S [48]; and a reduction in RMR is a known consequence of falling below one’s lower BF intervention point [47]. Most importantly, many of these physiological symptoms are overlapping outcomes of all three factors. Therefore, athletes should expect to experience the more consistently observed physiological adaptations given the demands of physique sport. Further, variables such as the degree of energy restriction, amount of body mass lost, length of diet, leanness relative to the lower BF intervention point of the individual, and approach to training will largely dictate when, and to what magnitude, an athlete undergoes these adaptations.

Physiological changes that were less consistent, such as changes in VO_2_ max, are also worthy of discussion. For example, while both overreaching and RED-S can degrade VO_2_ max, competitors who perform higher frequency or more intense AT sessions during contest preparation would be expected to experience some improvement in their aerobic fitness as well. Thus, the increase in AT would act as a competing positive signal to such declines in performance and potentially explain the divergent outcomes between individuals.

Finally, it is also worth mentioning that, while most competitors experienced reductions in RMR, RMR is but one component of total energy expenditure (TEE). Arguably, TEE is the most relevant factor when considering the necessary nutritional and AT adjustments a physique competitor must make across the competitive cycle. Further, determining whether a decrease in TEE occurred beyond what would be expected from weight loss is challenging based solely on changes in RMR from case studies, which can be confounded by acute factors such as changes in energy intake and expenditure, which occur regularly in athletes [49]. With that said, group-based research on physique athletes does indicate that reductions in RMR beyond what would be predicted by weight loss alone are common occurrences in both men and women [50]. However, one of the most variable aspects of TEE is non-exercise activity thermogenesis [51], changes to which may be of a larger magnitude than alterations in RMR in some cases [52]. Ultimately, future research is needed to comprehensively examine which components contribute to changes in TEE in response to contest preparation, and to what magnitude.

### 3.9. Psychometric Outcomes

Mood states (including perceived levels of stress) were assessed in 5 of the 11 included case series/case reports encompassing 5 male and 4 female athletes. There was considerable heterogeneity with the psychometric assessments used in the case reports, with no two studies using the same assessment. The inventories and scales used in the case studies were the State-Trait Anxiety Inventory, Perceived Stress Scale, Brunel Mood Scale, Activation Deactivation Adjective Checklist, and Profile of Mood States. There were no discernable sex-specific responses reported throughout contest preparation for mood states. Throughout most of the contest preparation periods, male and female athletes experienced low to moderate levels of perceived stress [16,44], and when anxiety levels were altered, they manifested as infrequent and intermittent elevations during contest preparation [5]. Two studies reported an increase in total mood disturbance as the competition date neared. In both case reports observing this effect, the increase in mood disturbance coincided with declines in energy and increases in tiredness/fatigue [20,21]. In both instances, total mood disturbances increased in the two months before competition.

Two case reports investigated self-reported assessments of eating behavior. One case report followed an experienced female figure athlete [23] and the other followed an experienced male bodybuilder [22]. Both reports used the same eating behavior questionnaire—the Three-Factor Eating Questionnaire-Revised 18-item version. The assessment relates to three different components of eating behavior: (1) cognitive restraint (conscious restriction of food for weight control); (2) uncontrolled eating (loss of control of food intake associated with hunger and overeating); and (3) emotional eating (the presence of irresistible emotional cues associated with eating). Both athletes reported similar eating behavior characteristics during their contest preparation periods for cognitive restraint. This subscale increased by ~40% during the competition phase for both athletes. In contrast, uncontrolled eating diverged between the male and female athletes. The male athlete reported an increase in uncontrolled eating from baseline to his competitive season, while the female athlete trended in the opposite direction, reporting a decrease in uncontrolled eating from baseline leading into her first competition.

A possible explanation for the differences in self-reported perceptions of uncontrolled eating between the male and female athletes may be the different changes in lean mass. The male athlete experienced a 3.4 kg reduction in lean mass during his contest preparation, while the female athlete experienced a slight increase in lean tissue (approximately 1 kg). Some studies suggest that a proportionate loss of lean mass during weight loss may influence appetite and energy intake [53]. Indeed, there are reports that the drive to eat is enhanced after lean tissue loss secondary to intentional weight loss [54]. It also is possible that BF levels may have played a role in this divergent outcome. The male athlete achieved a very low %BF (~5%), while the female athlete maintained a double-digit %BF pre-contest (~12%); even considering sex differences in essential fat levels, this could arguably be a meaningful difference. Given that leptin alters the drive to eat in response to changes in fat mass [19], it is conceivable that the effects of this hormone are magnified when BF levels reach a given lower threshold. However, Mäestu et al. [37] found that leptin levels decreased when BF levels fell from a mean of 9.6% to 8.0% during pre-contest preparation in a cohort of male bodybuilders, but did not decrease thereafter when BF levels further dropped to 6.5%, suggesting a functional lower limit for leptin levels. Further study is thus needed to test the veracity of this hypothesis as neither case study assessed leptin concentrations.

Finally, the two case reports assessing eating attitudes encompassed six athletes (three males and three females). Both reports used the Eating Attitudes Test^©^, which is a widely used standardized measure of symptoms and concerns characteristic of eating disorders. In both case reports across all athletes, there were no reported scores that would have prompted the athletes to seek the advice of a qualified mental health professional who has experience with treating eating disorders [5,16].

## 4. Limitations

Despite the robust insights obtained from the included cases studies, we must acknowledge limitations in drawing inferences from the data. Most notably, case study designs are observational in nature. While we can report on variables that seemed to change in concert across multiple case studies, causation cannot be determined. Further, all competitors were reportedly drug-free. However, several competitors self-reported to be free from PIEDs, but competed in non-tested competitions. Some of these competitors had their hormone levels tested as part of the case study, but others did not. Thus, it cannot be stated for certain that all competitors were in fact drug-free. In addition, if all competitors were drug-free, this presents an additional limitation as the present data may only be representative of drug-free competitors. It should also be noted that the questionnaires employed to assess mood and eating behavior are not specific to physique athletes and thus may not accurately reflect psychometric outcomes in the intended context. Lastly, all of the female competitors in the present review competed in the figure or bikini divisions, rather than the physique or bodybuilding divisions, which reward greater degrees of leanness and muscularity. Thus, it is possible that any divergent findings between men and women could be due to different divisional demands rather than sex alone.

## 5. Conclusions

Pre-contest preparation for physique competition elicits an array of chronic alterations in body composition, neuromuscular performance, chronic hormonal levels, physiological adaptations, and psychometric outcomes. Competitors consistently achieve substantial losses in BF across the pre-contest period, with male athletes ultimately achieving lower BF levels than women. Alternatively, female athletes tend to preserve lean mass to a greater extent compared with their male counterparts. Pre-contest preparation does not appear to have detrimental effects on bone mass, despite prolonged low energy availability.

The hormones testosterone, cortisol, and ghrelin are typically altered during contest preparation. Testosterone levels consistently dropped to below physiologically normal levels in the male athletes, but the levels were maintained or increased in the female athletes. Ghrelin and cortisol levels increased in most competitors as the contest preparation progressed. In contrast, the levels of estradiol and leptin were highly variable among the athletes.

Contest preparation is likely to result in absolute power decreases, with relative power in some cases remaining similar to pre-contest preparation levels. Competitors are also likely to experience absolute strength decreases in 1 RM strength, while maintaining, if not increasing, their relative 1 RM strength. When strength is assessed via other modalities (e.g., hand grip strength and knee extension strength), competitors may experience absolute and relative strength increases during their contest preparation period.

Physiological adaptations to pre-contest preparation including reductions in RMR, BP, and resting HR are commonplace among competitors. Further, among female competitors, disruption to and/or loss of the menstrual cycle is likely to occur at some stage of preparation. Sleep disruption is also reported, but insufficient data exist to determine how commonly this occurs, at what point, or to what severity. VO_2_ max may increase, decrease, or initially increase and then decrease depending on a combination of factors such as AT volume and intensity and the degree and timing of fatigue experienced during preparation. These physiological adaptations mirror many of the symptoms associated with RED-S, may also occur as a consequence of achieving extreme leanness, and are also sometimes observed in overreaching athletes, highlighting the multifactorial challenges of contest preparation.

Low to moderate perceived stress was experienced by some competitors during the competition preparation period. When present, such feelings were infrequent and intermittent. Not surprisingly, total mood disturbance (coinciding with decreases in energy and increases in tiredness/fatigue) scores increased closer to the bodybuilding competition dates. When measured, cognitive restraint (a subscale of eating behavior identifying the conscious restriction of food intake) was consistently elevated for both male and female competitors during the contest preparation phases. Moreover, there were no reports of eating behaviors that would have met the threshold for clinical characteristics of eating disorders.

## Figures and Tables

**Figure 1 jfmk-08-00059-f001:**
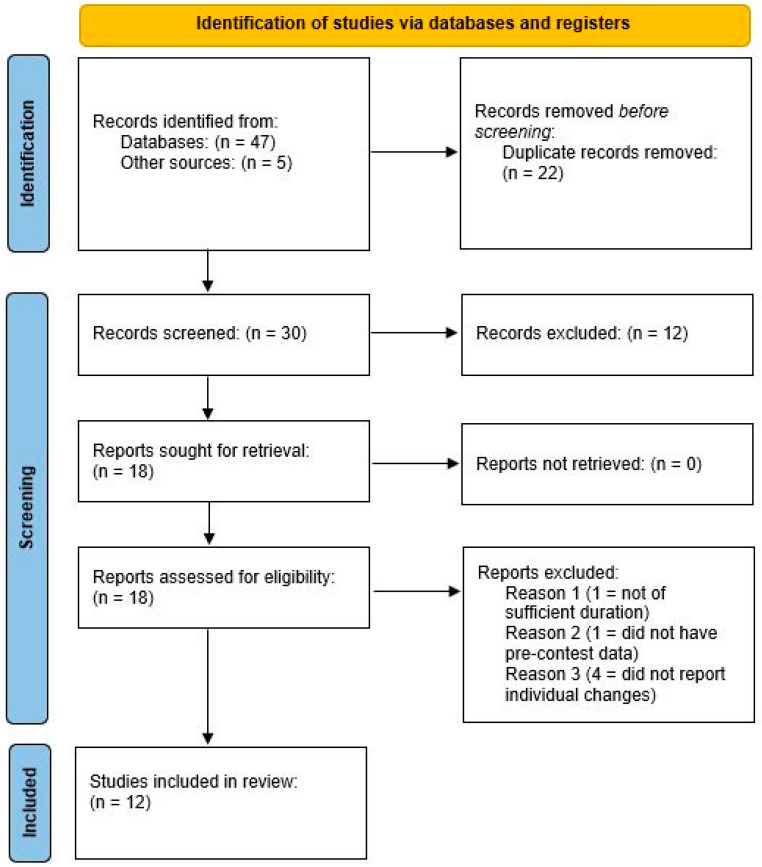
PRISMA flowchart of the search process.

**Table 1 jfmk-08-00059-t001:** Summary of the included studies.

Study	Sample	Duration	Measures
Chappell et al. [5]	(1) Drug-free, amateur male bodybuilder aged 30 yrs; (2) Drug-free, amateur male bodybuilder aged 44 yrs; (3) Drug-free, amateur male bodybuilder aged 38 yrs; (4) Drug-free, amateur female bodybuilder aged 38 yrs	26 wks	Body composition, neuromuscular, physiological, hormonal, psychometric
Halliday et al. [14]	Drug-free, amateur female figure competitor aged 26–27 yrs	20 wks	Body composition, physiological
Kistler et al. [15]	Drug-free, amateur male bodybuilder aged 26 yrs	26 wks	Body composition, physiological
Newmire et al. [16]	(1) Drug-free, amateur female bikini competitor aged 32 yrs; (2) Drug-free, amateur female bikini competitor aged 44 yrs	16 wks	Body composition, physiological, hormonal, psychometric
Pardue et al. [17]	Drug-free, amateur male bodybuilder aged 21 yrs	32 wks	Body composition, physiological, hormonal
Petrizzo et al. [18]	Drug-free, amateur female figure competitor aged 29 yrs	32 wks	Body composition
Robinson et al. [19]	Non-drug-tested amateur male physique competitor aged 26 yrs	14 wks	Body composition, physiological, neuromuscular
Rohrig et al. [20]	Drug-free, amateur female physique competitor aged 24 yrs	26 wks	Body composition, physiological, hormonal
Rossow et al. [21]	Drug-free, professional male bodybuilder aged 26–27 yrs	26 wks	Body composition, neuromuscular, physiological, hormonal, psychometric
Schoenfeld et al. [22]	Drug-free, amateur male bodybuilder aged 25 yrs	32 wks	Body composition, neuromuscular, physiological, hormonal, psychometric
Tinsley et al. [23]	Drug-free, amateur female figure competitor aged 27 yrs	25 wks	Body composition, neuromuscular, physiological, psychometric

## Data Availability

Not applicable.

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
