# Peer review of "Alterations in Measures of Body Composition, Neuromuscular Performance, Hormonal Levels, Physiological Adaptations, and Psychometric Outcomes during Preparation for Physique Competition: A Systematic Review of Case Studies"

_jfmk, 2023, doi:10.3390/jfmk8020059_

Round 1
Reviewer 1 Report
The reviewer congratulates the authors on putting together the manuscript. This is a valuable addition to the area. You’ve done an excellent job in pulling together relevant studies and summarising the data. The paper is also well written and I note no typo's or grammatical errors. I recommended that the manuscript be accepted by the journal. Most of the points I’ve raised are minor and the authors may wish to consider them to enhance the manuscript. For example I offer more modest reasons in some cases why a subject may have experienced a particular outcome.
I do believe though the paper would benefit from having a table or two to summarise the findings. This would help the reader in their understanding of the overall trends. This may prove difficult as there is likely to be missing data. It should be attempted though. A table summarising nutritional aspects for example noting dietary intakes including macronutrients, energy and relative intake, number of food items from the start to the end of the case studies, alongside exercise protocols would be helpful. While another which summaries physiological/hormonal/neuromuscular would be helpful.
Good work, I look forward to seeing the revised manuscript.
Introduction
Line 23: you may wish to note that the contest preparation phase or comp prep, is also referred to as the “on-season” or “in season”, in reference to the “off-season”. It seems logical since you referred to the “off-season”.
Line 28: The term performance and image enhancing drugs (PIEDs), is sometimes used. This might be useful in the context of this paper.
Line 36 to 39: This sentence is a little clunky, it might benefit from being broken into two separate sentences.
Results
Table 1: in the duration section, I believe it would be best to be consistent with the units if this is possible e.g. months vs weeks. It would be easier for the reader to interpret. Or you could include both units, e.g 26 weeks (6months) it looks like the table would have enough space.
The country where the case study was carried out would also be interesting. As would some sort of indication of how well trained or experienced the competitor is, Professional isn’t always an indictive of high level. If you went through the papers and were able to identify the events that all the competitors competed in (e.g natural or non-natural), this may be of value to you.
A legend might also be helpful for abbreviations e.g DF: drug free.
It’s worth mentioning in line 96 to 98, that one competitor (Robinson et al) is the only one to compete in a non-drug tested event.
Training and aerobic training
It would be useful if there were any sex differences
Discussion:
Line 168 to 170, essential fat in females is higher than compared to males. Leanness is relative in this respect e.g. 2.5% vs 10 to 12%. This is worth noting. Moreover, the appearance of low bodyfat and objective markers like a 7 point ISAK skinfolds from a female pre competition can be just as low as a male (sub 25 mm) despite the higher essential fat mass.
Do female athletes lose muscle mass at higher fat mass percentages consequently? Or because there is a higher essential fat threshold do, they only lose muscle mass once they start to lose essential fat? Are the female competitors really building muscle during this phase as potentially in experienced competitors or are the methods used to assess BF% lacking precision?
Line 179, I’m not sure you need to discuss the female athlete triad and the comparison with RED-S. You could simply explain RED-S and discuss it’s consequences in the context of bone health.
Hormonal levels
Line 200, It would be useful to know the range of testosterone values if you have this data, and how it compared at the start to the end of the trials. It would also be useful to see a comparison to reference values. If possible it would be useful to see these values for all hormones.
Neuromuscular Performance
Can you state the range of methods used to assess power e.g. wingate etc.
Did you note any changes in performance relative to athlete leanness? E.g. where athletes go beyond a leanness threshold relative strength/power is reduced. This would be worth mentioning.
Line 315: I believe it’s worth making a point about caloric energy thresholds around here. That once energy availability drops below a threshold hypothesised to be <25 kcal/kg bw, or relative energy availability drops below a threshold, this may attenuate normal physiological function in tandem with low body fat
Line 333, a graphic might be quite nice here to illustrate your point.
Line 358: It’s worth noting that many of the questionnaires used to assess eating attitudes, are not designed for bodybuilding populations. What might be considered as concerning behaviour in the general population are simply consequences of the comp prep phase and shouldn’t necessarily be considered an issue. This is worth noting. Moreover the increase in the male athletes uncontrolled eating might be less subtle, it may be brought on by disordered eating dietary with cheat meals or refeeding days since bodybuilders engage routinely in controlled eating.
Conclusions:
With regards to the female competitors the distinction between the classes is important, it’s worth stating that “at least in the so-called softer classes (bikini, figure) there may be etc. etc or that the athletes you investigated tended to follow x trend.
Reviewer 2 Report
This systematic review of case studies of athletes preparing for physique competitions is well described and organized. It provides a summary of the findings along the several areas that are well presented. This reviewer found no major issues with the manuscript as it addresses its primary purpose. One minor comment is for the inclusion criteria, make it clear all studies did not have to include all measurements being reviewed.
Author Response
This systematic review of case studies of athletes preparing for physique competitions is well described and organized. It provides a summary of the findings along the several areas that are well presented. This reviewer found no major issues with the manuscript as it addresses its primary purpose. One minor comment is for the inclusion criteria, make it clear all studies did not have to include all measurements being reviewed.
RESPONSE: Thank you for the effort in reviewing our paper. We appreciate the positive feedback. We have revised to state that all studies did not have to include all measures.